# Anthropogenic iron oxide aerosols enhance atmospheric heating

Nobuhiro Moteki[1], Kouji Adachi[2], Sho Ohata[1], Atsushi Yoshida[1], Tomoo Harigaya[1], Makoto Koike[1] & Yutaka Kondo[3]

Combustion-induced carbonaceous aerosols, particularly black carbon (BC) and brown carbon (BrC), have been largely considered as the only significant anthropogenic contributors to shortwave atmospheric heating. Natural iron oxide ($FeO_x$) has been recognized as an important contributor, but the potential contribution of anthropogenic $FeO_x$ is unknown. In this study, we quantify the abundance of $FeO_x$ over East Asia through aircraft measurements using a modified single-particle soot photometer. The majority of airborne $FeO_x$ particles in the continental outflows are of anthropogenic origin in the form of aggregated magnetite nanoparticles. The shortwave absorbing powers ($P_{abs}$) attributable to $FeO_x$ and to BC are calculated on the basis of their size-resolved mass concentrations and the mean $P_{abs}(FeO_x)/P_{abs}(BC)$ ratio in the continental outflows is estimated to be at least 4–7%. We demonstrate that in addition to carbonaceous aerosols the aggregate of magnetite nanoparticles is a significant anthropogenic contributor to shortwave atmospheric heating.

[1] Department of Earth and Planetary Science, Graduate School of Science, The University of Tokyo, Tokyo 113-0033, Japan. [2] Atmospheric Environment and Applied Meteorology Research Department, Meteorological Research Institute, Ibaraki 305-0052, Japan. [3] Arctic Environment Research Center, National Institute of Polar Research, Tokyo 190-8518, Japan. Correspondence and requests for materials should be addressed to N.M. (email: moteki@eps.s.u-tokyo.ac.jp).

Dark-coloured aerosols such as combustion-induced carbonaceous particles and wind-blown mineral dust absorb solar radiation and perturb the climate system by heating the atmosphere and reducing the snow albedo[1,2]. Shortwave atmospheric heating by such aerosols can alter the cloud fraction and hydrological cycle on both regional and global scales[3–6]. The global mean increase in precipitation per degree of increase in global-mean surface temperature, attributable to human-induced global warming, strongly depends on the emission of black carbon (BC), a proxy for anthropogenic shortwave absorbers[6]. To evaluate the effects of dark-coloured aerosols on climate using numerical models, observational datasets are needed to constrain the sources, atmospheric abundance, and detailed microphysical properties (for example, size distribution and morphology) of individual light-absorbing aerosols[1,7]. The long-term data on absorption aerosol optical depth measured by ground-based remote sensing[8] have been the main observational constraints for numerical models used to evaluate the effects of absorbing aerosols on climate change[1,7]. Inverting the remote sensing data to the column abundance of absorbing aerosols requires a priori assumption on the optical properties of each type of aerosols that contributes to absorption aerosol optical depth[9]. Therefore, in situ observations of all the important contributors to atmospheric shortwave absorption form the basis for the quantitative investigation of the complex effects of absorbing aerosols on climate.

Until now, two types of carbonaceous aerosols, BC and brown carbon (BrC), and wind-blown mineral dust (DU) have generally been considered as the only three aerosol species that contribute significantly to shortwave absorption in the atmosphere and snowpack[1,2,8,10,11]. BC and BrC are mostly co-emitted during the burning of fossil fuels, biofuels and biomass[7]. Light absorption by DU is mostly due to iron oxide nanoparticles attached on the surfaces of host mineral materials[12]. DU absorption dominates the atmospheric shortwave absorption over the desert and dust outflow regions[13]. A number of attempts have recently been made to quantify BrC absorption separately from BC absorption[11,14–16]. These studies suggest that BC absorption almost invariably dominates BrC absorption, except at ultraviolet wavelengths. In Beijing, an indirect estimate using ground-based remote-sensing technique[14] indicated that BrC absorption is ∼10% of BC absorption at the mid-visible wavelength in the winter and spring seasons. Another estimate of BrC absorption in Beijing using in situ optical and chemical measurements in March provided a similar result[11]. Using a global chemical transport model, Feng et al. estimated that BrC accounts for 7–19% of aerosol absorption (global average)[17].

Recent observational studies using electron microscopy methods revealed that anthropogenic iron oxide particles in the form of aggregated FeO$_x$ nanoparticles are ubiquitous in urban atmospheres[18–20] and roadside environments[21]. They are emitted from, for example, the blast furnaces of iron manufacturing facilities[22] and from the engines and brakes of motor vehicles[23,24]. The major iron oxide phases of aggregated FeO$_x$ nanoparticles are magnetite (Fe$_3$O$_4$)[18,20,21,23,24], maghemite (γ-Fe$_2$O$_3$)[21,24] and hematite (α-Fe$_2$O$_3$)[18,21,24]. However, their abundances in the troposphere and radiative effects have not yet been evaluated.

In this study, in situ aircraft measurements using a modified single-particle soot photometer (SP2)[25] and electron microscopy are performed to show that anthropogenic FeO$_x$ particles, particularly aggregated magnetite nanoparticles, are ubiquitous in the continental outflows from East Asia. We then evaluate their contribution to atmospheric shortwave absorption on the basis of the observed size-resolved mass concentrations and particle morphologies. Our results indicate that the absorption by anthropogenic FeO$_x$ is at least 4–7% of the BC absorption over East Asia.

## Results

**Observation.** We used observational data from the Aerosol Radiative Forcing in East Asia (A-FORCE) 2013W aircraft campaign[26] over Yellow Sea and East China Sea in February and March, 2013. Our modified SP2 can measure individual BC and FeO$_x$ particles in the mass-equivalent diameter ($D_m$) domains of $70\,nm \leq D_m \leq 850\,nm$ and $170\,nm \leq D_m \leq 2,100\,nm$, respectively[25]. The aerosol-sampling system, which consisted of a forward-facing shrouded solid diffuser inlet, transport tubes and aerosol measuring instruments, was designed to observe submicron-sized particles. The theoretical transmission efficiency curves Tr($D_m$) of FeO$_x$ particles began to drop at $D_m = \sim 600\,nm$ and decreased to $\sim 0.5$ at $D_m = 2,100\,nm$ (Supplementary Fig. 1). These theoretical Tr($D_m$) curves suggest that our measurement system underestimates the ambient FeO$_x$ concentrations at $D_m > \sim 600\,nm$.

**Characterization of anthropogenic FeO$_x$.** In this section, we characterize the fundamental properties of airborne FeO$_x$ particles in the East Asian continental outflows. For this purpose, we focus on the air parcel passed through the planetary boundary layer over eastern China and was directly transported to the sampling point on the flight track below 2 km altitude without experiencing wet removal of aerosols. The aircraft observation data of this air parcel is called dry PBL air[26]. The detailed criteria for selecting the dry PBL air according to the backward trajectory analysis is described in Kondo et al.[26].

First, we classified individual Fe-bearing particles in dry PBL air depending on the composition and morphology, based on the electron-microscopy analyses of 1,460 particles collected by an onboard aerosol-impactor sampler. A transmission electron microscope (TEM) and a scanning transmission electron microscope equipped with energy-dispersive X-ray spectrometry (STEM–EDS)[19] were used for these analyses. Table 1 lists the number of Fe-bearing particles for each of the three morphological types measured by the STEM–EDS analyses. The most abundant type of Fe-bearing particle was aggregated FeO$_x$ nanoparticles; the TEM images and associated elemental mappings of these

**Table 1 | Number of particles determined by STEM–EDS analysis for samples collected in dry PBL air.**

| Sample number | Flight date | Sampling time | All measured particles | Aggregate of FeO$_x$ nanoparticles | Fe-bearing fly ash | Fe-bearing mineral dust | Number fraction of mineral dust in all Fe-bearing particles |
|---|---|---|---|---|---|---|---|
| 1 | 4 Mar. | 14:35–14:47 | 278 | 20 | 6 | 1 | 0.038 |
| 2 | 7 Mar. | 11:53–12:05 | 344 | 4 | 4 | 0 | 0 |
| 3 | 7 Mar. | 12:53–13:05 | 403 | 11 | 2 | 0 | 0.10 |
| 4 | 8 Mar. | 11:17–11:29 | 435 | 1 | 9 | 1 | 0 |
| Total | | | 1460 | 36 | 21 | 2 | 0.035 |

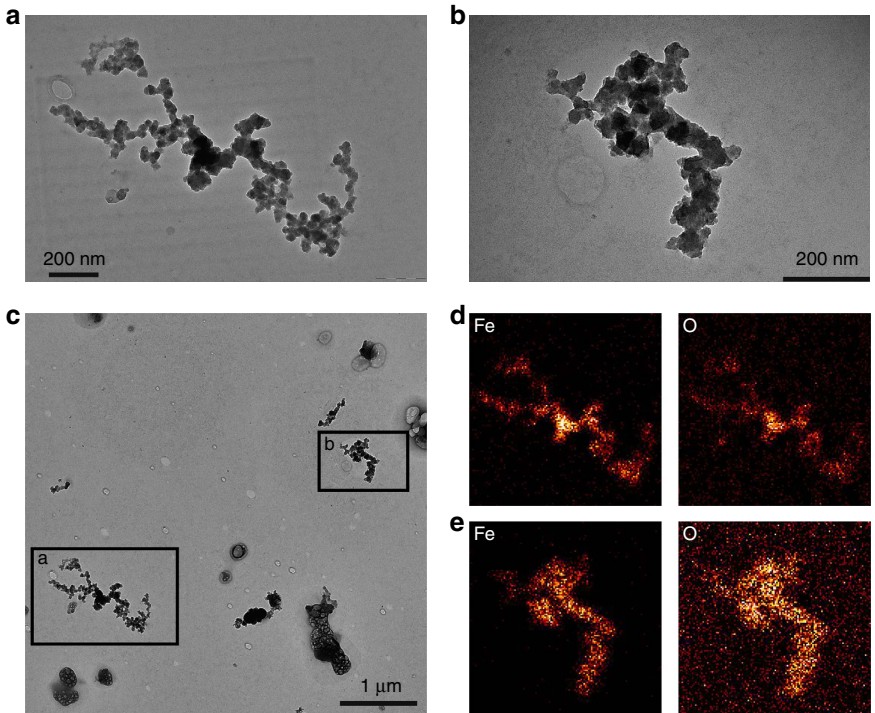

**Figure 1 | Transmission electron microscopy images and element distributions of aggregated iron oxide nanoparticles found in dry PBL air.** The sample was collected using an aerosol-impactor sampler onboard the aircraft from 14:35 to 14:47 on 4 March 2013 (local time) during the A-FORCE 2013W campaign (that is, Sample number 1 in Table 1). The particles magnified in **a** and **b** were collected on the same substrate with an inter-particle distance of ~4 μm, as shown in **c**. The elemental distributions of Fe and O for particles a and b are shown in **d** and **e**, respectively, and indicate that these particles contain iron oxide.

nanoparticles are shown in Fig. 1; Supplementary Fig. 2. The diameters of the $FeO_x$ monomers comprising each aggregate were highly variable, ranging from several nm to ~100 nm. These composition and morphology of $FeO_x$ particles are similar to those found in Tokyo[19] or Mexico City[27]. Our electron energy loss spectroscopy (EELS) analyses with TEM showed that the major component of the aggregated $FeO_x$ nanoparticles in dry PBL air is magnetite (Supplementary Fig. 3). The second most abundant type of Fe-bearing particles in dry PBL air was fly ash, which is a complex internal mixture of combustion-induced refractory materials commonly includes Si as a major component. The Fe-bearing fly ash particles in the dry PBL air typically contained only several per cent of Fe by mass. A third type of Fe-bearing particle, Fe-bearing mineral dust, was rare in the dry PBL air; the fraction of mineral dust particles among all the Fe-bearing particles was only 0.035 (Table 1). Although the mineral dust was observed to be minor in the dry PBL air, it would be the dominant type of Fe-bearing particles over the East Asia during an Aeolian dust (that is, Kosa) outflow event.

Next, we present results provided by the light-scattering signals of individual $FeO_x$ particles measured by the SP2. Figure 2 shows a scatterplot of the scattering cross-section at the onset of the incandescence ($C_{s\text{-oi}}$)[28] and the mass-equivalent diameter $D_m$ for both laboratory and ambient $FeO_x$-containing particles. Figure 2 also includes experimental results for pure magnetite particles and mineral dust particles (Icelandic dust and Taklamakan dust)[25]. Compared with pure magnetite, the mineral dust samples exhibited large $C_{s\text{-oi}}$ values beyond the detectable limit of ~$4 \times 10^{-14} \text{ m}^2$, because the host minerals internally mixed with the incandescing $FeO_x$ also contribute to the $C_{s\text{-oi}}$ value. On the basis of these results, we introduced a criterion for classifying the mixing state of detected $FeO_x$-containing particles in the $D_m$ domain of $170 \text{ nm} \le D_m \le 270 \text{ nm}$. Particles with $C_{s\text{-oi}} > 2 \times 10^{-14} \text{ m}^2$ were

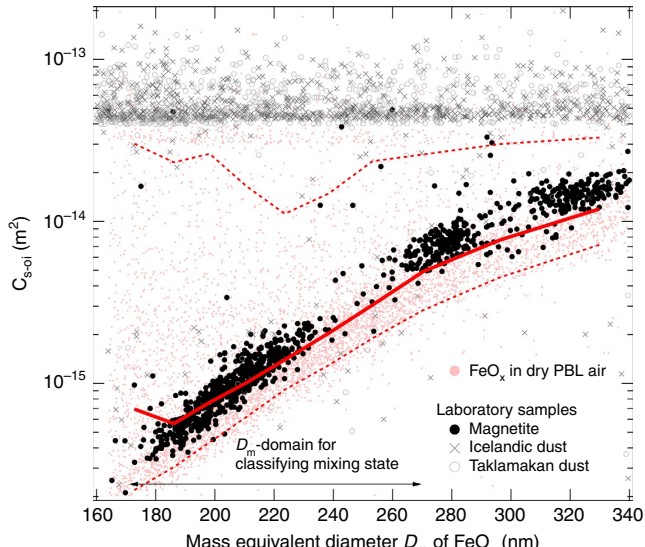

**Figure 2 | Single-particle soot photometer data indicating the mixing state of individual iron oxide-containing particles.** The figure shows scatterplots of scattering cross-section at the onset of incandescence ($C_{s\text{-oi}}$) and mass-equivalent diameter $D_m$ for laboratory samples (magnetite, Icelandic dust and Taklamakan dust) and $FeO_x$-containing particles in dry PBL air. For dry PBL data, the solid red line and two dashed red lines represent median and (10 and 90) percentile $C_{s\text{-oi}}$ values as functions of $D_m$, respectively. The $D_m$ domain used for classifying the mixing state of $FeO_x$-containing particles is schematically shown. In this study, ambient $FeO_x$-containing particles with $170 \text{ nm} \le D_m \le 270 \text{ nm}$ are classified as dust-like $FeO_x$ if $C_{s\text{-oi}} > 2 \times 10^{-14} \text{ m}^2$. The detectable upper limit of $C_{s\text{-oi}}$ is ~$4 \times 10^{-14} \text{ m}^2$.

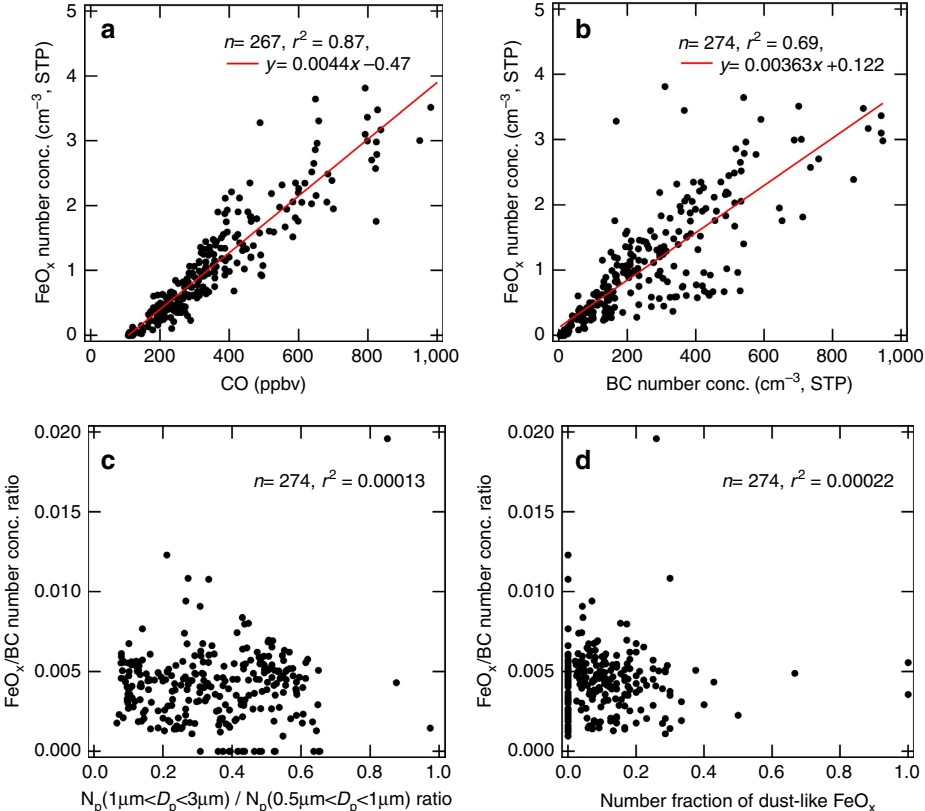

**Figure 3 | Scatterplots between the 1-min data of various observed parameters in dry PBL air.** Number of data points ($n$) and the square of the correlation coefficient ($r^2$) are shown in each panel. (**a,b**) The correlations between the $FeO_x$ concentrations and the concentrations of two representative pollutants in urban air: BC and CO. Particle concentrations were calculated at standard temperature and pressure (STP). The red line represents the linear regression line. (**c,d**) Scatterplots between the $FeO_x$/BC concentration ratio and two available indicators of relative abundance of mineral dust particles: (**c**) number concentration ratio of supermicron-sized aerosols to submicron-sized aerosols and (**d**) number fraction of dust-like $FeO_x$ particles. In **c**, $N_p$ denotes the number concentration of aerosols within a particular range of light-scattering equivalent diameter $D_p$, which was observed using a cloud and aerosol spectrometer probe.

classified as dust-like. Figure 2 shows that a majority of the $FeO_x$-containing particles in the dry PBL air were similar to the pure magnetite particles in terms of their $C_{s\text{-}oi}$ value, whereas a minority had substantially larger $C_{s\text{-}oi}$ values. In the dry PBL air, the fraction of dust-like $FeO_x$ particles among the detected $FeO_x$-containing particles was ~0.1, as discussed later.

In addition to the direct microscopic observations using the TEM–EELS, the optical signals measured using the SP2 also provided indirect information on the material properties of the airborne $FeO_x$. The timing of the onset of incandescence ($t_{oi}$) of $FeO_x$-containing particles has been experimentally shown to be a qualitative indicator of colour darkness (that is, light-absorbing efficiency) in $FeO_x$ materials[25]. The distribution of the $t_{oi}$ values of the $FeO_x$-containing particles in dry PBL air was similar to that for black-coloured magnetite but dissimilar to that of red-coloured hematite (see the Methods section and Supplementary Fig. 4). This implies that the major component of incandescing $FeO_x$ particles in dry PBL air is magnetite, which is consistent with the TEM–EELS results (Supplementary Fig. 3).

We found that the $FeO_x$ number concentration in the dry PBL air was highly correlated with the CO mixing ratio ($r^2 = 0.87$) and BC number concentration ($r^2 = 0.69$; Fig. 3a,b, respectively). These results suggest that the spatial distribution of the $FeO_x$ emission flux over the East Asian continent is similar to those of CO and BC. On the other hand, the $FeO_x$/BC number concentration ratio in the dry PBL air-correlated neither with the relative abundance of the supermicron-sized aerosols (Fig. 3c) nor the number fraction of the

dust-like $FeO_x$-containing particles (Fig. 3d). The scatterplots imply that the majority of the detected $FeO_x$ particles in the dry PBL air are not associated with DU, which is consistent with the results of the direct STEM–EDS analyses of the aerosol-impactor samples (Table 1).

The two most important conclusions drawn from these observations are as follows. First, the $FeO_x$ particles detected using the modified SP2, with the exception of the dust-like $FeO_x$, are primarily aggregated magnetite nanoparticles of anthropogenic origin. Second, the anthropogenic magnetite is a major type of Fe-bearing aerosol in East Asian continental outflow.

**Altitude profiles.** Figure 4 shows the size-resolved number and mass concentrations of $FeO_x$ at each altitude along with those in the dry PBL air. Within the observed size domain of $170\,\text{nm} \leq D_m \leq 2{,}100\,\text{nm}$, the size-resolved number concentration of $FeO_x$ was approximated using a power function with an offset

$$\left(\frac{dN}{d\log D_m}\right)_{FeO_x} = y_0 + aD_m^{-p}, \quad (1)$$

where $y_0$, $a$ and $p$ are numerical parameters listed in Table 2. The size-resolved mass concentration of $FeO_x$ was approximated as

$$\left(\frac{dM}{d\log D_m}\right)_{FeO_x} = \left(\frac{\pi}{6}\rho_{FeO_x}D_m^3\right)\left(\frac{dN}{d\log D_m}\right)_{FeO_x}, \quad (2)$$

where $\rho_{FeO_x}$ is the assumed bulk density of $FeO_x$ ($5.17\,\text{g cm}^{-3}$).

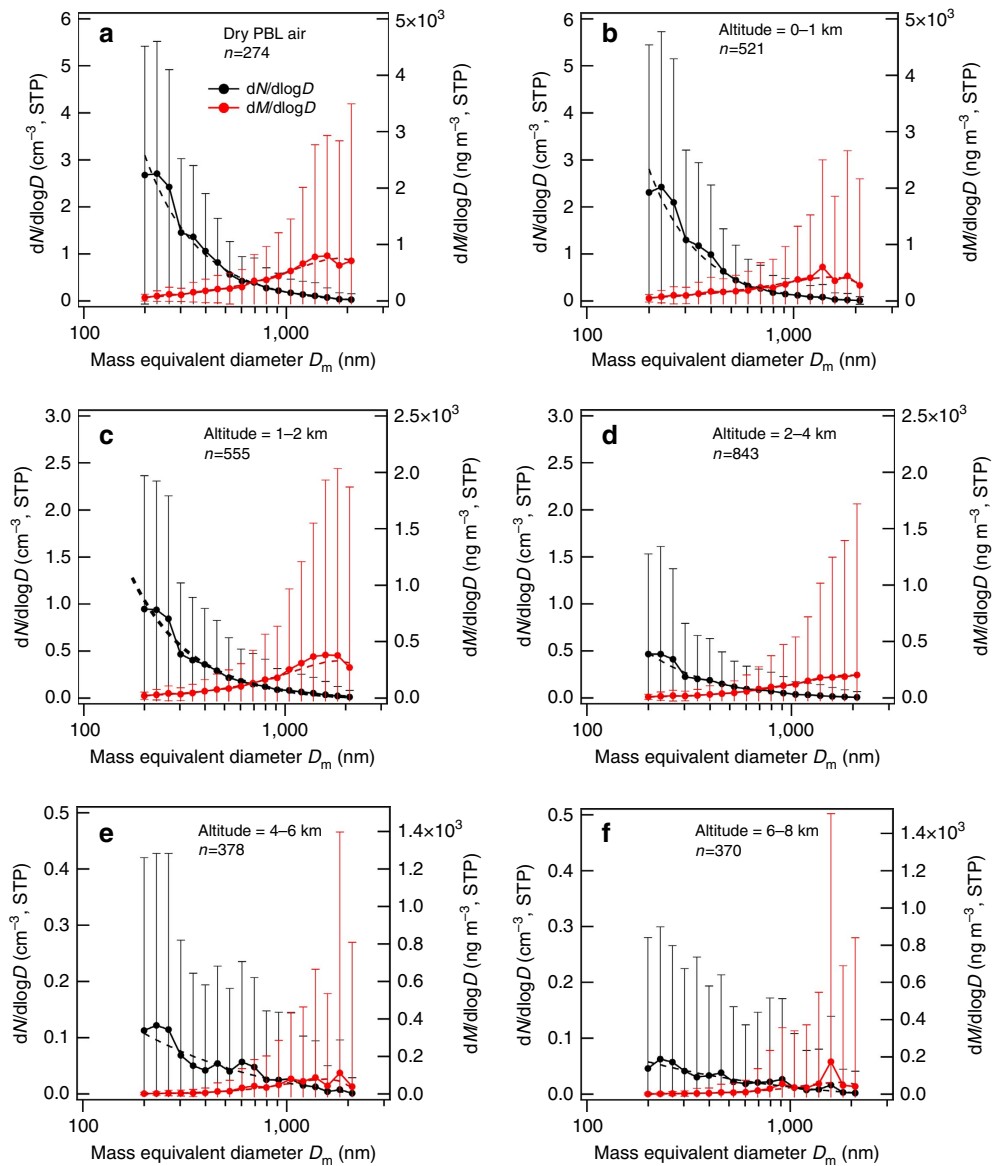

**Figure 4 | Size-resolved number and mass concentrations of iron oxide particles.** Results in dry PBL air and at different altitudes are shown in **a** and **b**–**f**, respectively. Filled circle and error bar represent the sample mean and the sample s.d. ($\pm 1\sigma$) of the 1-min data, respectively. The number of 1-min data points ($n$) is shown in each panel. Dashed lines show the results of least-square fitting of equation (1).

In contrast, the size-resolved mass concentration of BC within the observed size domain of $70\,\text{nm} \leq D_m \leq 850\,\text{nm}$ (see Figs 6 and 14 of Kondo *et al.*[26]) was approximated with a two-modal lognormal function as

$$\left(\frac{dM}{d\log D_m}\right)_{BC} = \sum_{i=0}^{1} A_i \exp\left[-\frac{\log^2(D_m/D_i)}{2\log^2\sigma_{gi}}\right], \qquad (3)$$

where $A_i$, $D_i$ and $\sigma_{gi}$ ($i = 0,1$) are numerical parameters listed in Table 2. Figure 4 shows that the mode $D_m$ values of the number size distribution were smaller than 170 nm, whereas the mode $D_m$ values of the mass size distribution may have been larger than 2,100 nm, that is, the detectable $D_m$ domain of $170\,\text{nm} \leq D_m \leq 2,100\,\text{nm}$ was insufficient to reveal the entire shape of the size-distribution function of ambient FeO$_x$. As mentioned earlier, our aerosol-sampling system likely underestimated the ambient FeO$_x$ concentrations for $D_m > {\sim}600\,\text{nm}$. In contrast, our BC measurements in the

domain of $70\,\text{nm} \leq D_m \leq 850\,\text{nm}$ were likely sufficient to determine the total BC mass concentrations, as expected from the Figure 6 and 14 of Kondo *et al.*[26] and the parameters listed in Table 2.

The BC size distributions tended to shift towards smaller sizes at higher altitudes[26] (see parameter $D_0$ in Table 2). This trend has been explained by the preferential wet removal of larger BC particles during vertical transport[26,29]. In contrast, the FeO$_x$ size distributions shifted towards larger sizes with increasing altitude (Fig. 4). The decrease of the parameter $p$ with altitude reflects this trend (Table 2). As shown later, the number fraction of dust-like FeO$_x$ was observed to increase with altitude. We hypothesize that the mineral dust particles lifted from the deserts in central China and transported by the westerlies cause the larger shift of the FeO$_x$ size distributions over the Yellow Sea and East China Sea.

Figure 5a,b show the vertical profiles of the FeO$_x$ concentrations and the FeO$_x$/BC concentration ratios, respectively. The

**Table 2 | List of the parameters for the size-distribution functions (equations (1) and (3)).**

| Data | $y_0$ | $a$ | $p$ | $A_0$ | $A_1$ | $D_0$ | $D_1$ | $\sigma_{g0}$ | $\sigma_{g1}$ |
|---|---|---|---|---|---|---|---|---|---|
| Dry PBL air | −0.03950 | 17340 | 1.627 | 1,648 | 538.9 | 187.3 | 433.4 | 1.495 | 1.949 |
| 0–1 km | −0.02937 | 38140 | 1.795 | 1,108 | 317.5 | 193.6 | 439.6 | 1.520 | 1.975 |
| 1–2 km | −0.02141 | 2488 | 1.465 | 559.4 | 168.7 | 183.8 | 454.0 | 1.509 | 2.200 |
| 2–4 km | −0.01328 | 585.0 | 1.336 | 524.9 | 124.6 | 180.2 | 319.7 | 1.466 | 2.158 |
| 4–6 km | −0.02993 | 3.874 | 0.6301 | 204.3 | 36.32 | 179.8 | 310.5 | 1.479 | 2.404 |
| 6–8 km | −0.02064 | 1.284 | 0.5271 | 78.28 | 12.87 | 178.0 | 528.8 | 1.580 | 1.359 |

Note: The physical units of $D_m$, $dN/dlogD_m$, and $dM/dlogD_m$ were assumed to be (nm), (cm$^{-3}$, STP) and (ng m$^{-3}$, STP), respectively.

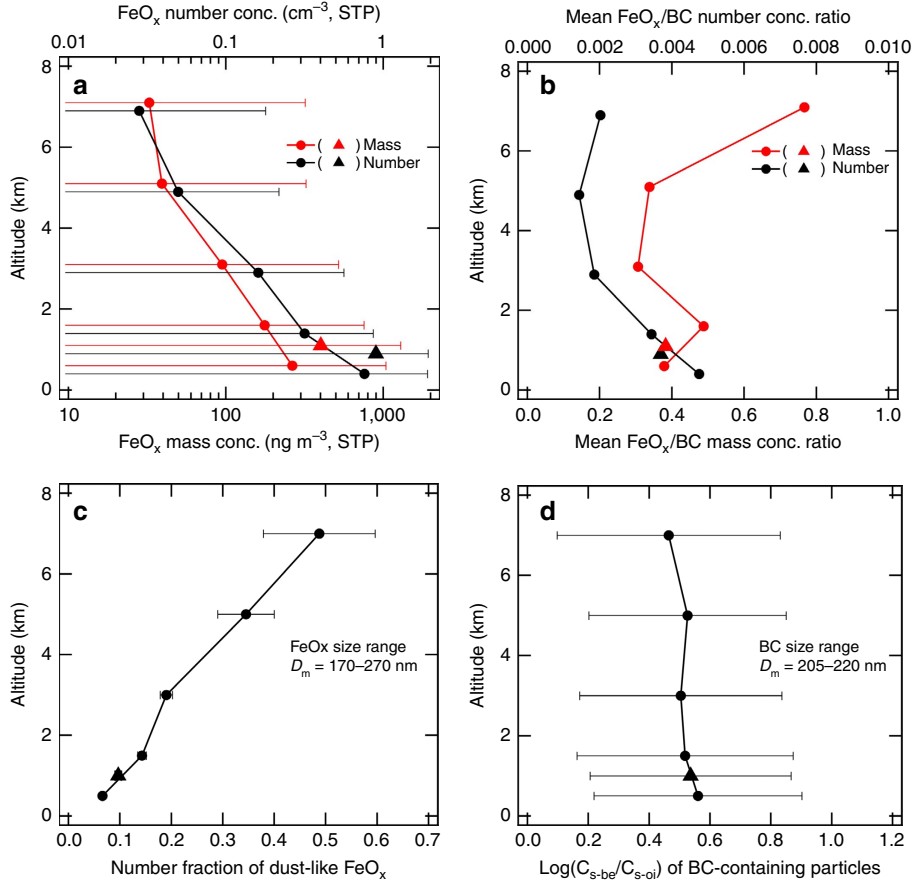

**Figure 5 | Altitude profiles of various observed parameters.** In each panel, the filled circles represent the mean values at each altitude, whereas the filled triangle represents the mean value in dry PBL air. (**a**) FeO$_x$ mass and number concentrations. The error bars in **a** represent the sample standard deviation (±1$\sigma$) of 1-min data. (**b**) The mean FeO$_x$/BC concentration ratios for mass and number. (**c**) The number fraction of dust-like FeO$_x$ for FeO$_x$ particle sizes in the $D_m$ domain of 170 nm ≤ $D_m$ ≤ 270 nm. The error bars in **c** represent uncertainty estimated by assuming that the number of particles detected in each altitude range follows a Poisson distribution. (**d**) The log($C_{s\text{-be}}/C_{s\text{-oi}}$) value of BC-containing particles for BC particle sizes in the $D_m$ domain of 205 nm ≤ $D_m$ ≤ 220 nm. The error bars in **d** represent the ±1$\sigma$ ranges of single-particle data.

mean FeO$_x$ mass concentration was 100–400 ng m$^{-3}$ at standard temperature and pressure (STP) in the boundary layer (altitude < ∼2 km), and monotonically decreased with altitude to 30–90 ng m$^{-3}$ STP in the free troposphere (altitude > ∼3 km). The mean FeO$_x$/BC number concentration ratio was 0.004 in the boundary layer and decreased to 0.002 in the free troposphere. The mean FeO$_x$/BC mass concentration ratio was 0.3–0.5 in all altitudes below 6 km, and increased to ∼0.8 at higher altitudes (6–8 km). The observed FeO$_x$/BC number (mass) ratio would be substantially larger if we were able to detect smaller (larger) FeO$_x$ particles with $D_m$ < 170 nm ($D_m$ > 2,100 nm). The number fraction of dust-like FeO$_x$ showed a monotonic increase with altitude from ∼0.07 at 0–1 km to ∼0.5 at 6–8 km (Fig. 5c). In an

aerosol-impactor sample collected during a time period of the highest FeO$_x$ number concentration observed at 6–8 km altitude (∼0.4 cm$^{-3}$ STP), we actually found mineral dust particles including Fe (Supplementary Fig. 5). However, in the free tropospheric samples, the number of Fe-bearing particles collected on each TEM grid was too low to evaluate the relative abundance of each of the three morphological types listed in Table 1.

**Mass-absorption cross-sections.** An original discrete-dipole approximation code was used to calculate the mass-absorption cross-sections of the BC- and FeO$_x$-containing particles

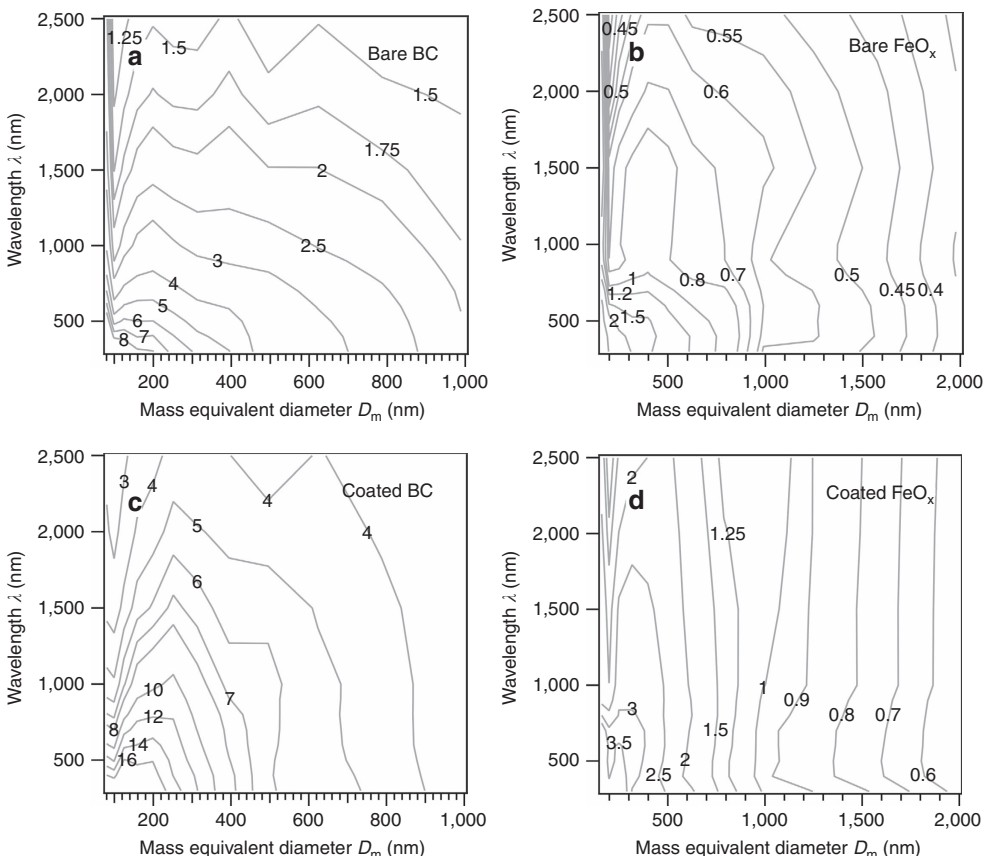

**Figure 6 | Mass-absorption cross-sections of black carbon- and iron oxide$_x$-containing particles as a function of wavelength and mass-equivalent diameter.** The mass-absorption cross-section $\sigma_a(\lambda, D_m)$ (m$^2$ g$^{-1}$) are computed for four distinct particle types: (**a**) bare BC, (**b**) bare FeO$_x$, (**c**) coated BC, and (**d**) coated FeO$_x$.

by assuming the particle shape, mixing state and refractive index of each material. Before discussing the results, we briefly explain the nontrivial assumptions used. For both BC and FeO$_x$, the particle shape was assumed to be a fractal-like aggregate of spherical monomers. The mixing state of the aggregate with other non-absorbing materials was assumed to be either bare or coated: the bare state denotes a pure aggregate, whereas the coated state denotes an aggregate coated by a large amount of non-absorbing material. In the coated state, the volume of the coating material was prescribed such that the theoretical log($C_{s\text{-be}}/C_{s\text{-oi}}$) value of the model BC-containing particles was greater than the measured log($C_{s\text{-be}}/C_{s\text{-oi}}$) values of the real BC-containing particles, where the $C_{s\text{-be}}$ denotes the scattering cross-section of a BC-containing particle in the SP2 laser beam before the onset of evaporation[28] (see the Methods section). The mean $+1\sigma$ value of the observed log($C_{s\text{-be}}/C_{s\text{-oi}}$) value for BC-containing particles with $D_m \sim 200$ nm was $\sim 0.9$ at all altitudes below 8 km (Fig. 5d). On the basis of this observation, the coating/aggregate volume ratio in the coated state was chosen to be approximately 3–4 so that the theoretical value of log($C_{s\text{-be}}/C_{s\text{-oi}}$) for the coated BC was $\sim 1.0$ at $D_m \sim 200$ nm (Supplementary Fig. 6).

The mass-absorption cross-sections ($\sigma_a$) calculated for the BC- and FeO$_x$-containing particles as functions of wavelength ($\lambda$) and mass-equivalent diameter $D_m$ are shown in Fig. 6. In general, the $\sigma_a$ value at a particular ($\lambda$, $D_m$) is approximately three times larger for BC than for FeO$_x$, primarily because the assumed bulk density of FeO$_x$ (5.17 g cm$^{-3}$) was 2.87 times greater than that of BC (1.8 g cm$^{-3}$). Compared with the bare state, $\sigma_a$ was

enhanced by a factor of approximately two in the coated state because of the so-called lensing effects[30]. The $\sigma_a(\lambda, D_m)$ of FeO$_x$ did not change appreciably with $\lambda$ in the $D_m$ domain largely contributing the total ambient FeO$_x$ mass concentration. Thus, we expect that the atmospheric absorption coefficient ($b_{abs}$) attributable to anthropogenic FeO$_x$ depends little on the wavelength. In contrast, the $b_{abs}$ attributable to carbonaceous aerosols (BC + BrC) is known to decrease sharply with wavelength[14,15,31]. This large difference in the wavelength dependence of $b_{abs}$ between anthropogenic FeO$_x$ and carbonaceous aerosols will be useful to classify them in remote sensing observations.

**Shortwave atmospheric heating.** Here we quantify the instantaneous shortwave atmospheric heating attributable to FeO$_x$ and BC using their observed size-resolved mass concentrations and theoretical mass-absorption cross-sections. The contribution to shortwave atmospheric heating from particles of a particular size is quantified by the size-resolved absorbing power d$P_{abs}$/dlog$D_m$ (W m$^{-3}$), which is defined as

$$\frac{\mathrm{d}P_{abs}}{\mathrm{dlog}D_m} = \int_{\lambda=250\text{nm}}^{2500\text{nm}} F_{ac}(\lambda)\sigma_a(\lambda, D_m)\left(\frac{\mathrm{d}M}{\mathrm{dlog}D_m}\right)\mathrm{d}\lambda, \quad (4)$$

where $F_{ac}(\lambda)$ is the spectral actinic flux (W m$^{-2}$ nm$^{-1}$), $\sigma_a(\lambda, D_m)$ is the mass-absorption cross-section (m$^2$ g$^{-1}$) and d$M$/dlog$D_m$ (g m$^{-3}$) is given by equations (2) or (3). The value of $F_{ac}(\lambda)$ for each altitude was computed using a radiative transfer model assuming a clear sky and a daytime-mean solar zenith

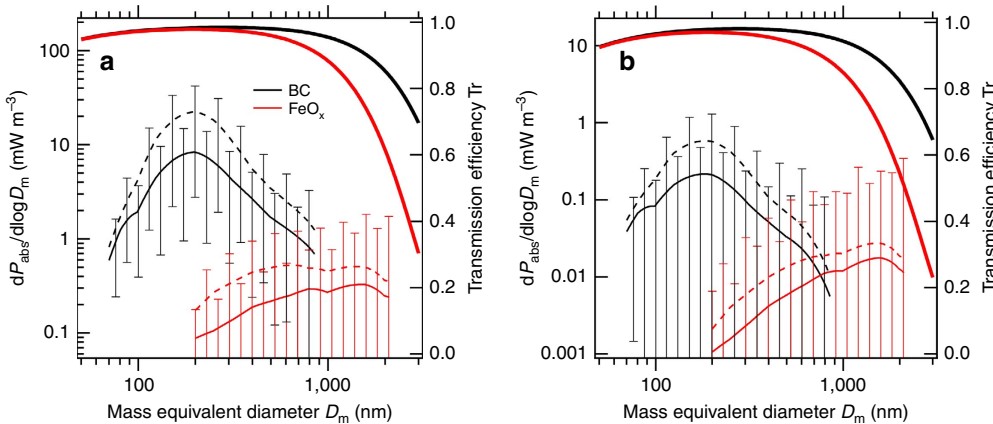

**Figure 7 | Size-resolved shortwave absorbing powers d$P_{abs}$/dlog$D_m$ of black carbon and iron oxide.** The d$P_{abs}$/dlog$D_m$ values (thin lines) were calculated for two distinct atmospheric conditions: (**a**) dry PBL air ($\sim$1 km altitude) and (**b**) the highest altitude (6–8 km). Solid and dashed lines represent the mean d$P_{abs}$/dlog$D_m$ values for bare and coated particles, respectively. Error bars represent the variability in the d$P_{abs}$/dlog$D_m$ values estimated from the $\pm 1\sigma$ value of the 1-min data of d$M$/dlog$D_m$. In each panel, the transmission efficiency curves Tr($D_m$) of the aerosol-sampling system calculated for BC and FeO$_x$ are also shown (thick lines).

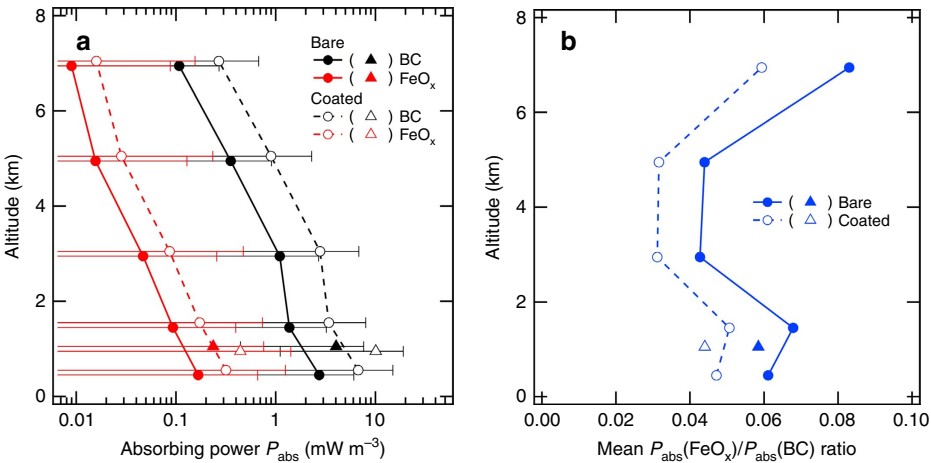

**Figure 8 | Altitude profiles of the shortwave absorbing powers $P_{abs}$ attributable to black carbon and iron oxide.** (**a**) The $P_{abs}$(BC) and $P_{abs}$(FeO$_x$), whereas **b** shows mean $P_{abs}$(FeO$_x$)/$P_{abs}$(BC) ratio. Filled and open circles (triangles) represent the mean values in each altitude range (dry PBL air) for the bare and coated particles, respectively. In **a**, error bars represent the variability in $P_{abs}$ estimated from the $\pm 1\sigma$ value of the 1-min mass concentration data.

angle (see the Methods section). Figure 7 shows the size-resolved absorbing powers of BC and FeO$_x$ in (a) dry PBL air and (b) at the highest altitude (6–8 km). The bell-shaped d$P_{abs}$/dlog$D_m$ distributions for BC suggest that the BC-containing particles within the detectable $D_m$ domain (70 nm $\leq D_m \leq$ 850 nm) predominantly contribute to the total BC absorption. On the other hand, the broad d$P_{abs}$/dlog$D_m$ distributions of FeO$_x$ suggest that the unobserved FeO$_x$-containing particles outside the detectable $D_m$ domain (170 nm $\leq D_m \leq$ 2,100 nm) also contribute significantly to the total FeO$_x$ absorption. Figure 7 also shows the theoretical transmission efficiency curves (Tr($D_m$)) of our aerosol-sampling system for BC and FeO$_x$ particles. The steeply decreasing Tr($D_m$) curve for FeO$_x$ suggests that we underestimated the d$P_{abs}$/dlog$D_m$ value for FeO$_x$ at $D_m > \sim$600 nm.

The total shortwave absorbing power $P_{abs}$ (W m$^{-3}$) was computed by integrating equation (4) over $D_m$:

$$P_{abs} = \int_{D_{min}}^{D_{max}} \left( \frac{dP_{abs}}{dlog D_m} \right) dD_m, \qquad (5)$$

where $D_{min}$ and $D_{max}$ denote the lower and upper limits of the

detectable $D_m$ domain, respectively. Figure 8a shows the altitude profiles of $P_{abs}$(BC) and $P_{abs}$(FeO$_x$). For both BC and FeO$_x$, $P_{abs}$ values decreased monotonically with altitude, following their mass concentration profiles. For the dry PBL air, the mean $P_{abs}$(BC) and $P_{abs}$(FeO$_x$) values in the bare (coated) state were 4.0 (10.0) and 0.24 (0.44) mW m$^{-3}$, respectively. These $P_{abs}$ values are equivalent to atmospheric heating rates of 0.29 (0.72) and 0.017 (0.032) K per day, respectively. Figure 8b shows the altitude profiles of the mean $P_{abs}$(FeO$_x$)/$P_{abs}$(BC) ratio. In the boundary layer wherein anthropogenic FeO$_x$ dominated the detected FeO$_x$ particles (Fig. 5c), the mean $P_{abs}$(FeO$_x$)/$P_{abs}$(BC) ratio was estimated to be 0.04–0.07. The altitude dependence of $P_{abs}$(FeO$_x$)/$P_{abs}$(BC) was similar to that of the FeO$_x$/BC mass concentration ratio (Fig. 5b). In the highest altitude (6–8 km), wherein the dust-like FeO$_x$ particles contributed 50% of the detected FeO$_x$ particles (Fig. 5c), the $P_{abs}$ attributable to anthropogenic FeO$_x$ may be $\sim$50% of the calculated $P_{abs}$ value.

It should be noted that the $P_{abs}$(FeO$_x$)/$P_{abs}$(BC) ratios reported herein are lower than the real values because our measurements underestimated the ambient FeO$_x$ mass concentration in the $D_m$ domain of $\sim$600 nm $\leq D_m \leq$ 2,100 nm and totally ignored

the $FeO_x$ particles outside the detectable $D_m$ domain. Considering these facts, the real $P_{abs}(FeO_x)/P_{abs}(BC)$ ratios in the boundary layer would be as large as 0.1.

## Discussion

On the basis of the previous observations of BrC near Beijing in March 2013 (refs 11,14), we roughly estimate the typical $P_{abs}(BrC)/P_{abs}(BC)$ value in the boundary layer to be $\sim 0.1$. Thus, we expect that $P_{abs}(FeO_x)$ is as large as $P_{abs}(BrC)$ in the East Asian continental outflows. In future studies, it is highly desirable to use an aerosol-sampling system with a higher transmission efficiency of large $FeO_x$ particles. In addition, some technical improvements in the SP2 are necessary to perform $FeO_x$ measurements beyond the current detectable $D_m$ domain (170 nm $\leq D_m \leq$ 2,100 nm). Despite the room for such improvements, an essential conclusion can be drawn from our results, namely, in addition to BC and BrC, airborne $FeO_x$ in the form of aggregated magnetite nanoparticles should also be recognized as a significant anthropogenic contributor to shortwave atmospheric heating.

In addition to clear-sky shortwave absorption, we briefly discuss some other potential climate effects of anthropogenic $FeO_x$ particles. Low stratiform clouds are of frequent climatological occurrences over the mid- to high-latitude ocean and southeast China[32]. Under modest maximum supersaturation in such clouds, particle's mass fraction activated to cloud droplets will be substantially larger for $FeO_x$ than BC, because the critical supersaturations of $FeO_x$-containing aerosols will be lower than those of BC due to the larger $D_m$. Under these situations, $P_{abs}(FeO_x)/P_{abs}(BC)$ is enhanced because the lensing effect in droplets[33] is larger for $FeO_x$ than for BC. We expect that the number of co-emitted cloud condensation nuclei and their precursor gases in anthropogenic $FeO_x$-rich sources, which remain uncertain, are substantially different from those in anthropogenic BC- and BrC-rich sources such as residential coal, industrial coal and biomass fuels[7]. Thus, without comprehensive investigations, it is not clear whether the net positive climate forcing of anthropogenic $FeO_x$-rich sources is negligible or comparable with those of BC- and BrC-rich sources. Finally, it should be mentioned that the anthropogenic $FeO_x$ particles may also play a role in the biogeochemical cycles[34].

## Methods

**Modified single-particle soot photometer.** A modified single-particle soot photometer (SP2), which detects light-absorbing refractory aerosols on the basis of intra-cavity laser-induced incandescence, was used to measure BC and iron oxide $FeO_x$ particles[25]. The BC and $FeO_x$ are discriminated from each other on the basis of the colour ratio, which is an indicator of the boiling temperature[25], and the peak amplitude of the blue-band incandescence signal, which is an indicator of an incandescing particle's size[25]. Supplementary Figure 7 shows a scatterplot of the peak amplitudes and colour ratios of all incandescing particles detected during the A-FORCE 2013 W campaign. The boundary lines for discriminating $FeO_x$ from BC were determined on the basis of experimental results[25] and are also shown in Supplementary Fig. 7.

The masses ($m$) of the individual $FeO_x$ and BC particles were determined from the peak heights of the incandescence signals using the experimentally determined mass-to-peak height relationships[25]. The mass-equivalent diameters ($D_m$) for BC and $FeO_x$ were calculated from the observed masses $m$ assuming the void-free densities of 1.8 g cm$^{-3}$ and 5.17 g cm$^{-3}$ (density of magnetite), respectively. In this study, the detectable size domains of the BC and $FeO_x$ particles were 70 nm $\leq D_m \leq$ 850 nm and 170 nm $\leq D_m \leq$ 2100 nm, respectively.

The detailed morphological properties of the individual particles containing a particular mass of each incandescing material were evaluated on the basis of time-resolved scattering cross-sections $C_s$ (integrated over the solid angle of light collection) in a laser beam derived from the scattering signal[35–37]. We evaluated whether the incandescing material (BC or $FeO_x$) was attached to the surface of another particle on the basis of the magnitude of $C_s$ at the onset of the incandescence signal ($C_{s-oi}$)[28]. In our observations, the fraction of the attached-type[28] BC-containing particles with $D_m = 200$ nm was $<3\%$ at any altitudes below 8 km. Thus, we assumed the $C_{s-oi}$ of BC-containing particles to be equivalent to the $C_s$ of BC core. For BC-containing particles, the ratio of $C_s$ before the onset of

particle evaporation ($C_{s-be}$) to $C_{s-oi}$ is used as an indicator of the amount of non-refractory material (for example, sulfate) coating the BC[28].

The timing of the onset of the incandescence signal ($t_{oi}$) is an indicator of the heating rate of an absorbing particle in the laser beam of the SP2; $t_{oi}$ tends to be lower (earlier) for more efficient light absorbers. Our experiments[25] showed that the $t_{oi}$ of magnetite ($Fe_3O_4$) particles was markedly earlier than that of hematite ($\alpha$-$Fe_2O_3$), reflecting the greater absorption efficiency of magnetite. It should be noted that comparisons of $t_{oi}$ values derived from the laboratory experiments and field data are meaningful only under similar SP2 conditions (that is, the same laser power and width of the Gaussian beam) because the $t_{oi}$ of a particular composition depends on these parameters. Supplementary Fig. 4 shows the $t_{oi}$ distributions for the ambient BC and $FeO_x$ particles in dry PBL air. The SP2 laser power during the A-FORCE 2013W campaign was similar to that in our laboratory experiments[25], as expected from the comparison of $t_{oi} - t_{cen}$ (0.2 µs) for BC between ambient data ($-55 < t_{oi} - t_{cen} < -30$) and our laboratory results ($-50 < t_{oi} - t_{cen} < -35$; refer to Fig. 2c of Yoshida et al.[25]). The $t_{oi}$ distributions of the $FeO_x$ particles for two different mass ranges (10 fg $\leq m \leq$ 88 fg and $m > 530$ fg) were similar, indicating that the primary $FeO_x$ material does not change appreciably with $FeO_x$ particle mass (Supplementary Fig. 4). Our experiment illustrated that the incandescing probability of pure hematite particles was zero for $m < \sim 100$ fg because the absorption efficiency of hematite particles in this size range is insufficient to heat the particles to the incandescing temperature[25]. Thus, the incandescing $FeO_x$ particles in dry PBL air had greater absorption efficiency than hematite. The $t_{oi} - t_{cen}$ (0.2 µs) distributions of $FeO_x$ with 10 fg $\leq m \leq$ 88 fg and $m > 530$ fg largely overlapped with the $t_{oi} - t_{cen}$ distributions of pure magnetite with $m < \sim 100$ fg ($-40 < t_{oi} - t_{cen} < -20$)[25] and $m > 530$ fg ($-40 < t_{oi} - t_{cen} < -30$)[25], respectively. On the basis of these results and the EELS spectra shown in Supplementary Fig. 3, the major constituent of the detected $FeO_x$ particles in dry PBL air is likely magnetite ($Fe_3O_4$).

**Electron microscopy analysis.** An aerosol impactor-sampler[19] onboard the aircraft was used to collect aerosol samples on the Cu TEM grids with collodion substrate at 12-min intervals during each flight. A 120-kV transmission electron microscope (JEM-1400, JEOL) equipped with an energy-dispersive X-ray spectrometer (Oxford Instruments) was used for the STEM–EDS analysis. A 200-kV transmission electron microscope (ARM 200, JEOL) was used for the EELS analysis.

**Transmission efficiency of aerosols.** Special care is required to measure the number concentrations of supermicron-sized aerosols in ambient air as the transmission efficiency (Tr) through the entire tubing apparatus connecting the aerosol inlet to the particle detection volume unavoidably decreased because of the inertial and gravitational depositions. Since a substantially large fraction of the total $FeO_x$ mass was expected to be in the supermicron size range (Fig. 4), our conclusion is strongly dependent on the degree of sampling loss of large $FeO_x$ particles. According to the theoretical formulae compiled by Pramod et al.[38], we estimated the $Tr(D_m)$ curves of the BC and $FeO_x$ particles through our aerosol-sampling system. In our A-FORCE 2013W aircraft campaign, the sample air was aspirated through a forward-facing shrouded inlet (DMT Inc., Boulder, CO, USA) installed on the top of the aircraft fuselage. This aerosol inlet is a replica of 'University of Hawaii shrouded solid diffuser inlet' described and evaluated by McNaughton[39]. An isokinetic aspiration was performed to maintain the aspiration efficiency at $\sim 1.0$, independent of the particle size. The geometric specifications and flow rates of the tubing apparatuses comprising the sampling system are listed in Supplementary Table 1. The void-free densities of BC and $FeO_x$ were assumed to be 1.8 and 5.17 g cm$^{-3}$, respectively. The dynamic shape factors of the BC and $FeO_x$ particles were assumed to be 1.5. Our experiment using pure magnetite particles confirmed the reasonable agreement between the measured and theoretical $Tr(D_m)$ curves for a $\frac{1}{4}$-inch tube (length $=$ 0–3 m) assuming the same density and dynamic shape factor.

**Computing the fractal-like aggregates of spheres.** On the basis of the TEM observations of ambient BC- and $FeO_x$-particles, we assumed the shapes of the model BC- and $FeO_x$-particles for the electromagnetic scattering calculations to be fractal-like aggregates of spheres. The aggregate geometry was computed using an original tunable cluster–cluster aggregation (CCA) code called aggregate_gen, which is an efficient C++ implementation of the hierarchical CCA algorithm[40]. We assumed the fractal prefactor and fractal dimension to be $k_f = 1.0$ and $D_f = 2.8$, respectively. The monomer diameters of BC and $FeO_x$ were assumed to be 40 and 80 nm, respectively. The number of monomers $N_{pp}$ in the aggregates ranged from 8 to 16384. This range of $N_{pp}$ covers the observed $D_m$ domain for both BC and $FeO_x$.

**Electromagnetic scattering calculations.** The mass-absorption cross-sections for BC and $FeO_x$ as functions of both wavelength $\lambda$ and mass-equivalent diameter $D_m$ were computed using an original electromagnetic scattering solver called block-DDA, which utilizes the block Krylov subspace methods[41–43] to efficiently solve the discrete-dipole approximation (DDA)[44,45] for multiple incident waves, or equivalently, for multiple target orientations. For every ($\lambda$, $D_m$) condition, we used the mean value of the results from four different randomly chosen target

orientations. When applying block-DDA to the scattering problems for fractal-like aggregates of BC or $FeO_x$ monomers, the monomer dipole formulation was assumed to avoid the discretization shape error in each monomer[46]. The coupled electric and magnetic dipoles formulation[46,47] was used to mitigate the multipole-truncation error associated with the monomer-dipole assumption[46]. To apply efficient fast Fourier transform-based algorithms in DDA[48], we adjusted the center position of each monomer dipole to the nearest site of the computational cubic lattice (CL)[46]. In this study, the lattice spacing of the CL was set to one-half of the monomer diameter. The geometry of the coating material surrounding a fractal-like aggregate of spheres was computed by the following procedure. First, the CL sites in the proximity of each monomer (at a distance of less than $T$ monomer radii) were selected as candidates for coating volume elements, where the coating thickness was controlled by the parameter $T$. In this study, $T$ was selected to be 3. Next, the CL sites overlapping with monomer volume, if present, were removed from the candidates. Finally, the coating volume was assigned to each candidate.

We used the complex refractive indices of BC and $FeO_x$ listed in Supplementary Table 2. Ackerman and Toon[49] also used this refractive index data set to estimate the radiative effects of atmospheric aerosols containing BC and magnetite. The refractive index of the coating material was assumed to be $1.5 + 0.0i$, independent of the wavelength.

**Radiative transfer calculations.** The spectral actinic fluxes $F_{ac}(\lambda)$ under the A-FORCE 2013W condition were calculated using the radiative transfer software package libRadtran (version 1.6 beta)[50]. At each altitude, the absorption coefficients $b_{abs}(\lambda)$ $(m^{-1})$ of the BC and $FeO_x$ particles were calculated on the basis of observational $D_m/dlogD_m$ data and theoretical $\sigma_a(\lambda, D_m)$ values. The extinction coefficients $b_{ext}(\lambda)$ were evaluated as $b_{abs}(\lambda)/(1 - \omega)$, where the single-scattering albedo $\omega$ was assumed to be 0.85, independent of $\lambda$. The Henyey–Greenstein function with asymmetry parameter 0.7 was used for the scattering phase function. Shortwave absorption by gases (for example, water vapour and ozone) was calculated using the LOWTRAN/SBDART parameterization assuming the default atmospheric profiles (US-standard atmosphere). Effective solar zenith angle averaged over the daylight hours (local noon ± 6 h) was assumed for the actinic flux calculation. All the input parameters used in the radiative transfer calculations are listed in Supplementary Table 3.

**Cloud and aerosol spectrometer probe.** A cloud and aerosol spectrometer probe (CAS; Droplet Measurement Technologies, Inc.) installed under a major wing of the aircraft was used for measuring size-resolved number concentration of total aerosol in the light-scattering diameter $(D_p)$ domain of $0.5\,\mu m \leq D_p \leq 50\,\mu m$. The CAS instrument used herein was also described in Koike et al.[51].

**Code availability.** The aggregate_gen code for computing the fractal-like aggregates of spheres and the block-DDA code for electromagnetic scattering were developed by the corresponding author (N.M.) and are available in the GitHub repository at: https://github.com/nmoteki.

**Data availability.** $FeO_x$ data shown in Fig. 4 is available at NOAA National Center for Environmental Information (NCEI) Reference ID: YKJKWD. The other observational data and calculation results are available from N. Moteki.

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

## Acknowledgements

This work was supported by the Ministry of Education, Culture, Sports, Science, and Technology (MEXT), the global environment research fund of the Japanese Ministry of the Environment (A-1101, 2-1403 and 5-1605), the Japan Society for the Promotion of Science (JSPS) KAKENHI Grants 15H05465, 16K16188 and 15H02811, the GRENE Arctic Climate Change Research Project, and the Arctic Challenge for Sustainability (ArCS) project. The TEM–EELS study was supported by the NIMS microstructural characterization platform as a program of the Nanotechnology Platform of MEXT, Japan. We thank Diamond Air Service Inc. for supporting the A-FORCE campaign. We thank N. Takegawa, K. Kita, H. Matsui, N. Oshima, T. Mori and H. Hashioka for supporting the observations.

## Author contributions

N.M. conceived the idea and wrote the paper. N.M., M.K. and K.A. acquired the SP2 data, CAS data and samples for electron microscopy analysis during the aircraft campaign. T.H., S.O. and A.Y. analysed the observational data. A.Y. and S.O. performed the laboratory experiments for the SP2 characterization using magnetite and dust samples. K.A. performed the STEM–EDX and TEM–EELS analyses. N.M. performed the calculations of aerosol transmission efficiency $Tr(D_m)$, electromagnetic scattering and radiative transfer. A.Y. and N.M. experimentally tested the accuracy of the theoretically calculated $Tr(D_m)$ curves. M.K. and Y.K. proposed the A-FORCE campaigns.

## Additional information

**Competing interests:** The authors declare no competing financial interests.

