## [Peer Review File · Nature Communications]

Reviewers' comments:

Reviewer #1 (Remarks to the Author):

This is a very interesting, important, and well-written manuscript arguing the importance of anthropogenic iron oxides for total ambient aerosol absorption and radiative heating. It should be published in Nature Comm. After the following comments have been taken into account:

1. L94: "lacked any correlation": Please quantify this comment by giving the correlation coefficient R^2 for Fig. 1b, just as you give it for Fig. 1a.
2. L169-171: "The essential quantity of interest is the $P_{abs}(FeOx)/P_{abs}(rBC)$ ratio, indicating the relative contribution of FeOx to the shortwave atmospheric absorption compared to that of the well-studied rBC." I would argue that this is not the "essential quantity of interest" as we are interested in the contribution of (a) aerosol to Top of Atmosphere (TOA) heating and (b) the vertical distribution of this heating. TOA heating is determined by two intrinsic aerosol parameters, the single scattering albedo (SSA) and the upscatter fraction, one extrinsic aerosol parameter, the aerosol optical depth (AOD) and by the albedo of the underlying scene (i.e., cloud or surface albedo) (Chylek and Wong 1995). Therefore, the question if aerosols heat or cool the earth system is determined by their SSA and upscatter fraction as function of the underlying albedo and the authors should estimate these quantities for the FeOx and rBC. Of special importance is that rBC particles are much smaller than FeOx particles and therefore have very different phase functions and upscatter fractions. As the authors already have done detailed numerical calculation of particle optical properties, this should be quite doable. An example for different aerosols can be found in Chakrabarty et al. (2014).
3. L188-190: "the magnitude of AAOD attributed to anthropogenic iron oxide is not always significantly smaller than that of black carbon and may be comparable to those of mineral dust and brown carbon." The comparison with mineral dust and brown carbon should be quantified (back of the envelope estimation) or deleted.
4. Error estimates are often absent with values in the text and in tables generally missing error estimates and figures missing error bars.

REFERENCES

Chakrabarty, R. K., N. D. Beres, H. Moosmüller, S. China, C. Mazzoleni, M. K. Dubey, L. Liu and M. I. Mishchenko (2014). "Superaggregates from Flaming Wildfires and Their Direct Radiative Forcing." Nature Scientific Reports 4: doi:10.1038/srep05508.
Chylek, P. and J. Wong (1995). "Effect of Absorbing Aerosol on Global Radiation Budget." Geophys. Res. Lett. 22(8): 929-931.

Reviewer #2 (Remarks to the Author):

Review of "Anthropogenic iron oxide aerosols enhance atmospheric heating" by Nobuhiro Moteki et al.

In the manuscript "Anthropogenic iron oxide aerosols enhance atmospheric heating", the authors use a modified SP instrument to measure light absorbing aerosols based on laser-induced incandescence. Their analysis of color ratios of the incandescence signals reveals a distinct distribution of iron-containing absorbing aerosols. Based on a lack of a correlation of the fraction of large aerosol and the fraction of iron containing particles they conclude that it would be "contradictory if the majority of detected FeOx particles were the free iron component of mineral dust". However, in my view the presented evidence is too weak for the strong conclusions drawn,

attributing the signal of FeOx entirely to “anthropogenic” iron oxides. Overall, I was missing a critical view on the results and the associated uncertainties, combined with a tendency over-emphasize potential effects. I can therefore not recommend publication of this manuscript in Nature Communications.

Major issues

1. The conclusion that the lack of a correlation of the fraction of large aerosol and the fraction of iron containing particles would be “contradictory if the majority of detected FeOx particles were the free iron component of mineral dust” appears very strong, in particular in the light of the observed large size distribution of iron oxides, fully compatible with components associated with mineral dust. This should be seen in the light of an underestimation of the reported large particle sizes due to measurement limitations, without which the observed size distribution would be even more consistent with mineral dust, and their own work (Yoshida et al., 2016), showing that the observed color ratios are fully consistent with mineral dust.

2. The conclusion that “Based on the observational evidence obtained from our STEM-EDS, TEM-EELS, and SP2 analyses as well as the previous reports mentioned above, the FeOx particles detected in the CH-outflow and TK-urban were likely the aggregates of magnetite (Fe₃O₄) nanoparticles.” is insufficiently backed up by evidence. We are presented with a few selected TEM images but it is entirely unclear how representative these particles are for the total number of particles sampled or how the selections was made.

3. Line 138: “The FeOx/rBC concentration ratio in CH-outflow allowed us to estimate the emission flux of anthropogenic FeOx in eastern China as the emission flux of rBC in this region has been observationally constrained.” This is another example of a strong statement, ignoring the uncertainties in the underlying analysis. 1. It is far from trivial to constrain emission fluxes observationally.

Responses to Reviewer's comments to the manuscript:

We thank two reviewers for giving critical comments useful to improve the quality of our manuscript. In this revision, new experimental and theoretical results are included to strengthen the robustness of our conclusion. We summarize major and minor points in this revision before giving responses to individual comments.

Major revisions:

1. Observation data

In this revision, we use all data from the A-FORCE 2013 W aircraft campaign for showing broader results including altitude profiles. For succinctness, data from the ground observation in Tokyo are not included in the revised manuscript. Because of the omission of “TK-urban” dataset, the “CH-outflow” dataset was renamed to “dry PBL air” (i.e., the original name given by Kondo et al. 2016 JGR).

2. TEM analysis of FeOx particles

We performed additional STEM-EDS analyses to provide statistical data of Fe-bearing particles observed in dry PBL air. Number of particle counts for total aerosols, aggregated FeOx nanoparticles, Fe-bearing fly ash, and Fe-bearing mineral dust are shown in Table 1. These results confirm that airborne FeOx in dry PBL air are mostly of anthropogenic origin (i.e., aggregated FeOx nanoparticles or Fe-bearing fly ash). Additional TEM-EELS analyses of many-different aggregated FeOx nanoparticles provided additional evidences of that major phase of iron oxide in these particles is magnetite (Supplementary Fig. 3).

3. Identification of dust-like FeOx particles using the SP2 signals

We have performed additional laboratory experiment using mineral dust samples (Icelandic dust and Taklamakan dust) for characterizing the SP2 scattering signals for Fe-bearing dust particles. We empirically found that the scattering cross-section at the onset of incandescence C_{s-oi} can be used for discriminating the dust-like FeOx particles (i.e., internal mixture of FeOx with huge amount of refractory mineral material) from the pure FeOx particles (e.g., aggregated FeOx spheres) (see Fig. 2). In this revision, number fraction of dust-like FeOx particles is used as an indicator of relative abundance of mineral dust particles.

4. Data correction of size-resolved FeOx concentrations

We have revised the procedure for discriminating the FeOx from BC using the incandescence signals. In old procedure, the threshold “color ratio” is assumed to be constant independent of the peak height of incandescence signal. In new procedure, the threshold color ratio slightly changes as a function of signal's peak height to avoid potential misclassifications (Supplementary Fig. 7). The new procedure corrects unintentional overestimate of FeOx concentrations in supermicron D_m -domain. Because of this data correction, FeOx mass concentration has been reduced about 20% from the old value.

5. Some corrections in statistical results

Before revision, we used the median value of 1-minute data for showing the FeOx concentration and FeOx/BC ratio in each altitude level (old Table 1). We have noticed that median FeOx mass concentration is biased by large Poisson noise of 1-minute data: Number of FeOx particles detected during 1-minute interval were usually zero and occasionally 1 or 2 in higher altitudes. For this reason, the median value of 1-minute data in higher altitudes is zero which is systematically lower than the mean value. To avoid this lower bias, we use the mean value instead of the median value (Fig 5). In revised manuscript, altitude profiles of FeOx concentration and FeOx/BC ratio have been substantially altered by this correction.

Minor revisions:

1. Error bars

Error bars indicating the result's variability (or uncertainty) are shown in each figure (Fig. 4, 5, 7, and 8). Definition of "error" is explained in the figure caption.

2. Calculations of particle's mass absorption cross-sections

We have revised the assumptions and computational methods for estimating the mass absorption cross-sections in following points. (1) Aggregate's monomer diameters assumed for BC (50 nm) and FeOx (200 nm) have been reduced to 40 nm and 80 nm, respectively. The revised values are more-or-less similar to the real monomer sizes. (2) Variability of the mass absorption cross-section attributable to the internal-mixing with non-absorbing compounds (e.g., sulfate) is also evaluated: We performed computations assuming the two-distinct mixing states "bare" and "coated" (see Fig. 6). A discrete-dipole approximation technique designed for fractal-like aggregate of spheres was used for these computations (see Methods section).

Responses to individual comments

Reviewer #1 (Remarks to the Author):

This is a very interesting, important, and well-written manuscript arguing the importance of anthropogenic iron oxides for total ambient aerosol absorption and radiative heating. It should be published in Nature Comm. After the following comments have been taken into account:

Answer:

Thank you for positive comments.

1. L94: "lacked any correlation": Please quantify this comment by giving the correlation coefficient R^2 for Fig. 1b, just as you give it for Fig. 1a.

Answer:

Following your suggestion, we include the r_2 value in every scatterplot.

2. L169-171: "The essential quantity of interest is the $P_{\text{abs}}(\text{FeOx})/P_{\text{abs}}(\text{rBC})$ ratio, indicating the relative contribution of FeOx to the shortwave atmospheric absorption compared to that of the well-studied rBC." I would argue that this is not the "essential quantity of interest" as we are interested in the contribution of (a) aerosol to Top of Atmosphere (TOA) heating and (b) the vertical distribution of this heating. TOA heating is determined by two intrinsic aerosol parameters, the single scattering albedo (SSA) and the upscatter fraction, one extrinsic aerosol parameter, the aerosol optical depth (AOD) and by the albedo of the underlying scene (i.e., cloud or surface albedo) (Chýlek and Wong 1995). Therefore, the question if aerosols heat or cool the earth system is determined by their SSA and upscatter fraction as function of the underlying albedo and the authors should estimate these quantities for the FeOx and rBC. Of special importance is that rBC particles are much smaller than FeOx particles and therefore have very different phase functions and upscatter fractions. As the authors already have done detailed numerical calculation of particle optical properties, this should be quite doable. An example for different aerosols can be found in Chakrabarty et al. (2014).

Answer:

Following your suggestion, the revised manuscript includes the vertical distributions of $P_{\text{abs}}(\text{BC})$, $P_{\text{abs}}(\text{FeOx})$ in addition to the $P_{\text{abs}}(\text{FeOx})/P_{\text{abs}}(\text{BC})$ ratio (Fig. 8). Because our work is a case study based on in situ aircraft data obtained at particular locations and periods, the magnitude of P_{abs} (and TOA heating as well) estimated herein would not be of fundamental quantity which will be useful in other situations. For this reason, the main focus of our discussion in the revised manuscript is still on the $P_{\text{abs}}(\text{FeOx})/P_{\text{abs}}(\text{BC})$ ratio, which is almost insensitive to the variability of aerosol's concentration and actinic flux.

3. L188-190: "the magnitude of AAOD attributed to anthropogenic iron oxide is not always significantly smaller than that of black carbon and may be comparable to those of mineral dust and brown carbon." The comparison with mineral dust and brown carbon should be quantified (back of the envelope estimation) or deleted.

Answer:

Following your critical suggestion, we deleted these statements.

4. Error estimates are often absent with values in the text and in tables generally missing error estimates and figures missing error bars.

Answer:

Following your comment, we have included error bars in each figure. Please see minor revision 1.

REFERENCES

Chakrabarty, R. K., N. D. Beres, H. Moosmüller, S. China, C. Mazzoleni, M. K. Dubey, L. Liu and M. I. Mishchenko (2014). "Superaggregates from Flaming Wildfires and Their Direct Radiative Forcing." *Nature Scientific Reports* 4: doi:10.1038/srep05508.
Chýlek, P. and J. Wong (1995). "Effect of Absorbing Aerosol on Global Radiation Budget." *Geophys. Res. Lett.* 22(8): 929-931.

Answer:

Thank you for introducing these papers. I have found them very interesting.

Reviewer #2 (Remarks to the Author):

Review of "Anthropogenic iron oxide aerosols enhance atmospheric heating" by Nobuhiro Moteki et al.

In the manuscript "Anthropogenic iron oxide aerosols enhance atmospheric heating", the authors use a modified SP instrument to measure light absorbing aerosols based on laser-induced incandescence. Their analysis of color ratios of the incandescence signals reveals a distinct distribution of iron-containing absorbing aerosols. Based on a lack of a correlation of the fraction of large aerosol and the fraction of iron containing particles they conclude that it would be "contradictory if the majority of detected FeOx particles were the free iron component of mineral dust". However, in my view the presented evidence is too weak for the strong conclusions drawn, attributing the signal of FeOx entirely to "anthropogenic" iron oxides. Overall, I was missing a critical view on the results and the associated uncertainties, combined with a tendency over-emphasize potential effects. I can therefore not recommend publication of this manuscript in *Nature Communications*.

Answer:

Thank you for carefully reading our manuscript and giving critical comments. In this revision, we have additional observational evidences which strengthen the reliability of our conclusions. We believe that your suggestions are carefully incorporated in the revised manuscript.

Major issues

1. The conclusion that the lack of a correlation of the fraction of large aerosol and the fraction of iron containing particles would be "contradictory if the majority of detected FeOx particles were the free iron component of mineral dust" appears very strong, in particular in the light of the observed large size distribution of iron oxides, fully compatible with components associated with mineral dust.

Answer:

In this revision, we have included additional observational evidences showing that the FeOx particles in East Asian outflow are mostly anthropogenic magnetite-like nanoparticles (e.g., Table 1). Please see major revision 2-3.

This should be seen in the light of an underestimation of the reported large particle sizes due to measurement limitations, without which the observed size distribution would be even

more consistent with mineral dust, and their own work (Yoshida et al., 2016), showing that the observed color ratios are fully consistent with mineral dust.

Answer:

In this revision, “fraction of large aerosol” is derived using the data from the cloud and aerosol spectrometer probe (CAS) installed outside the fuselage instead of data from UHSAS. This avoids potential artifacts from particle’s transmission efficiency of aerosol sampling system.

2. The conclusion that “Based on the observational evidence obtained from our STEM-EDS, TEM-EELS, and SP2 analyses as well as the previous reports mentioned above, the FeOx particles detected in the CH-outflow and TK-urban were likely the aggregates of magnetite (Fe₃O₄) nanoparticles.” is insufficiently backed up by evidence. We are presented with a few selected TEM images but it is entirely unclear how representative these particles are for the total number of particles sampled or how the selections was made.

Answer:

In this revision, we have performed additional STEM-EDS analyses for several different aerosol-impactor samples and provided number of counts for each type of Fe-bearing particles (Table 1). We also included additional TEM-EELS results in Supplementary Fig 3. Please see major revision 2. In addition, analysis of particle’s mixing state using the SP2 scattering signal enables discrimination of dust-like FeOx particles (internally-mixed with other mineral particles) from pure FeOx particles (please see Fig. 2). Please see major revision 3. These additional results confirm that FeOx particles detected by the SP2 in East Asian outflow are mostly aggregate of magnetite nanoparticles.

3. Line 138: “The FeOx/rBC concentration ratio in CH-outflow allowed us to estimate the emission flux of anthropogenic FeOx in eastern China as the emission flux of rBC in this region has been observationally constrained.” This is another example of a strong statement, ignoring the uncertainties in the underlying analysis. 1. It is far from trivial to constrain emission fluxes observationally.

Answer:

Following your critical comment, we have deleted statements about emission fluxes.

REVIEWERS' COMMENTS:

Reviewer #1 (Remarks to the Author):

As stated in my review of the previous version, this is a very interesting, important, and well-written manuscript arguing the importance of anthropogenic iron oxides for total ambient aerosol absorption and radiative heating. The authors have responded to my previous comments to my satisfaction and this manuscript should be published in Nature Communications.

Reviewer #3 (Remarks to the Author):

- Title and abstract are clear.
- I think the author have demonstrated their conclusions.

Minor comment:

I found the section on health at the end (lines 318-327) totally out of context. By contrast, I would move that - perhaps as a short sentence - in the introduction. There is plenty of evidence in the literature (cited already in the manuscript) showing urban ultrafine aerosol rich in iron. To support that, you can also use ref 33 in the introduction. For example, in lines 76-82, when citing ref 18-20, add ref 33 and complete the introduction section.

In one of my recent work with the University of Birmingham (showing submicron iron particles from urban environment) we also believe these submicron urban iron containing aerosol may also play a role in the biogeochemical cycles (Fine Iron Aerosols Are Internally Mixed with Nitrate in the Urban European Atmosphere. Dall'Osto M., et al. Environmental Science and Technology, 50, 4212-4220. DOI: 10.1021/acs.est.6b01127), and may differ across different continents (i.e. Asia and Europe)

Reviewer #4 (Remarks to the Author):

This is a very interesting manuscript that contains important new information. I think it should eventually be published. It needs some attention to details and polishing first. I provide a list of detailed comments below.

p. 5, line 98 – I would another sentence here saying if the nozzle was a razor edge or curved leading edge. This makes quite a difference in transmission of aerosols through the inlet. Also, I would add here that it was surrounded by a shroud. Include the diameter of the transmission tube. Several are listed in Table 1 (supplemental) but I could not determine which size was actually employed. The smaller the diameter the poorer the transmission (determined empirically).

p. 5, line 100 – remember this was estimated from theory, not actual testing which would likely reduce the transmission more.

p. 5, line 105 – explain what you mean by “dry PBL air”. Little precipitation? (line 107) I have never heard of this terminology. Why was it only encountered below 2 km altitude? The authors seem to emphasize the importance of this dry PBL air, but why?

p. 6, line 111 – were the aerosol impactor samples collected on the ground or in the air? This is the first time this type of sampling was mentioned.

What type of filter media was employed in collecting all of the aerosol samples? I don't see where

this is mentioned in the text.

p. 6, line 127 – the dust was low at this time of year, but in the springtime it could be the dominant form of the iron aerosols over China. You might want to point this out.

p. 7, line 147 – where were these impactor samples collected?

p. 7, line 160-162 – again, this may be dependent on the season of the year. I would soften this statement.

p. 10, line 195 – the iron oxide shifted to larger size as altitude increased. To me this suggests that dust might be the source of this aerosol. We know that it is lifted from deserts in central China and is transported eastward over the Pacific. I would make this more explicit of a statement than is given in line 199.

p. 11, line 214 – I would re-word this to make it read clearer.

Responses to Reviewer's comments to the manuscript:

We greatly thank three reviewers for reading our manuscript carefully and giving useful comments to improve the perspective and readability of the manuscript. Please find our point-by-point responses to reviewer's comment below.

Reviewer #1 (Remarks to the Author):

As stated in my review of the previous version, this is a very interesting, important, and well-written manuscript arguing the importance of anthropogenic iron oxides for total ambient aerosol absorption and radiative heating. The authors have responded to my previous comments to my satisfaction and this manuscript should be published in Nature Communications.

Answer:

Thank you for reading the manuscript again and giving an encouraging comment.

Reviewer #3 (Remarks to the Author):

- Title and abstract are clear.
- I think the author have demonstrated their conclusions.

Minor comment:

I found the section on health at the end (lines 318-327) totally out of context. By contrast, I would move that - perhaps as a short sentence - in the introduction. There is plenty of evidence in the literature (cited already in the manuscript) showing urban ultrafine aerosol rich in iron. To support that, you can also use ref 33 in the introduction. For example, in lines 76-82, when citing ref 18-20, add ref 33 and complete the introduction section.

Answer:

Following your suggestion, we have removed the paragraph on health effect in the discussion section and moved the reference Maher et al. 2016 to the introductory section, as one of the known evidence of presence of FeOx aerosols in urban air.

In one of my recent work with the University of Birmingham (showing submicron iron particles from urban environment) we also believe these submicron urban iron containing aerosol may also play a role in the biogeochemical cycles (Fine Iron Aerosols Are Internally Mixed with Nitrate in the Urban European Atmosphere. Dall'Osto M., et al. Environmental Science and Technology, 50, 4212-4220. DOI: 10.1021/acs.est.6b01127), and may differ across different continents (i.e. Asia and Europe)

Answer:

Thank you for a comment which broaden the perspective. We have shortly mentioned this point at the end of discussion section: “Finally, it should be mentioned that the anthropogenic FeO_x particles may also play a role in the biogeochemical cycles³⁴”.

Reviewer #4 (Remarks to the Author):

This is a very interesting manuscript that contains important new information. I think it should eventually be published. It needs some attention to details and polishing first. I provide a list of detailed comments below.

p. 5, line 98 – I would another sentence here saying if the nozzle was a razor edge or curved leading edge. This makes quite a difference in transmission of aerosols through the inlet. Also, I would add here that it was surrounded by a shroud. Include the diameter of the transmission tube. Several are listed in Table 1 (supplemental) but I could not determine which size was actually employed. The smaller the diameter the poorer the transmission (determined empirically).

Answer:

As you suggested, we have detailed the description of the shrouded solid diffuser inlet employed in our aircraft measurements in Methods section (Transmission efficiency of aerosols). Especially, we have added a reference McNaughton et al. [2007] in which the characteristics of the same type of inlet was fully described.

p. 5, line 100 – remember this was estimated from theory, not actual testing which would likely reduce the transmission more.

Answer:

Following your comment, the sentence has been reworded to clarify that the Tr curves are estimated from theory: “These theoretical $\text{Tr}(D_m)$ curves suggest that our measurement system underestimates the ambient FeO_x concentrations at $D_m > \sim 600$ nm.”

p. 5, line 105 – explain what you mean by “dry PBL air”. Little precipitation? (line 107) I have never heard of this terminology. Why was it only encountered below 2 km altitude?

The authors seem to emphasize the importance of this dry PBL air, but why?

Answer:

Thank you for this useful comment for improving the readability. We have reworded the descriptions of the “dry PBL air” dataset to clarify the definition and purpose. The dry PBL air was focused to investigate the emission characteristics avoiding the effects of wet removal. The corresponding sentences have been revised to: “For this purpose, we focus on the “dry PBL air” dataset²⁶, which corresponding to the air parcel passed through the planetary boundary layer over eastern China and was directly transported to the sampling point on the flight track below 2 km altitude without experiencing wet removal of aerosols. The detailed criteria for selecting the dry PBL air according to the backward trajectory analysis were described in Kondo et al.²⁶.”

p. 6, line 111 – were the aerosol impactor samples collected on the ground or in the air? This is the first time this type of sampling was mentioned.

Answer:

Surely these samples are collected in the air (using an onboard aerosol-impactor sampler). We have clarified this point by rewriting the sentences: “First, we classified individual Fe-bearing particles in dry PBL air depending on the composition and morphology, based on the electron-microscopy analyses of 1460 particles collected by an onboard aerosol-impactor sampler.”

What type of filter media was employed in collecting all of the aerosol samples? I don't see where this is mentioned in the text.

Answer:

We employed the Cu TEM grids with collodion substrate for collecting all of the aerosol samples. The Method section (Electron microscopy analysis) has been revised to clarify this point.

p. 6, line 127 – the dust was low at this time of year, but in the springtime it could be the dominant form of the iron aerosols over China. You might want to point this out.

Answer:

Following your suggestion, we have pointed out that the mineral dust can be a dominant type of Fe-bearing particles over the East Asia during an Aeolian dust outflow event: “Although the mineral dust was observed to be minor in the dry PBL air, it would be the dominant type of Fe-bearing particles over the East Asia during an Aeolian dust (i.e., Kosa) outflow event.”

p. 7, line 147 – where were these impactor samples collected?

Answer:

Sorry for ambiguous writing. We have reworded this sentence to clarify our intention herein: “In addition to the direct microscopic observations using the TEM-EELS, the optical signals measured using the SP2 also provided indirect information on the material properties of the airborne FeO_x.”

p. 7, line 160-162 – again, this may be dependent on the season of the year. I would soften this statement.

Answer:

We have added the words “in the dry PBL air” in line 195, to stress that the observation results in Figure 3c,d may not be generalized to all the seasons: “On the other hand, the FeO_x/BC number concentration ratio in the dry PBL air correlated neither with the relative abundance of the supermicron-sized aerosols (Fig. 3c) nor the number fraction of the dust-like FeO_x -containing particles (Fig. 3d).”

p. 10, line 195 – the iron oxide shifted to larger size as altitude increased. To me this suggests that dust might be the source of this aerosol. We know that it is lifted from deserts in central China and is transported eastward over the Pacific. I would make this more explicit of a statement than is given in line 199.

Answer:

Following your suggestion, we have revised the sentence to give a more explicit interpretation of the altitude dependence of FeO_x size distribution: “We hypothesize that the mineral dust particles lifted from the deserts in central China and transported by the westerlies cause the larger shift of the FeO_x size distributions over the Yellow and East China seas.”

p. 11, line 214 – I would re-word this to make it read clearer.

Answer:

Thank you for an advise for improving the readability, we have reworded the sentence to be simpler: “However, in the free tropospheric samples, the number of Fe-bearing particles collected on each TEM grid was too low to evaluate the relative abundance of each of the three morphological types listed in Table 1.”